# Brain Tumor Characterization Using Radiogenomics in Artificial Intelligence Framework

**DOI:** 10.3390/cancers14164052

**Published:** 2022-08-22

**Authors:** Biswajit Jena, Sanjay Saxena, Gopal Krishna Nayak, Antonella Balestrieri, Neha Gupta, Narinder N. Khanna, John R. Laird, Manudeep K. Kalra, Mostafa M. Fouda, Luca Saba, Jasjit S. Suri

**Affiliations:** 1Department of CSE, International Institute of Information Technology, Bhubaneswar 751003, India; 2Department of Radiology, AOU, University of Cagliari, 09124 Cagliari, Italy; 3Department of IT, Bharati Vidyapeeth’s College of Engineering, New Delhi 110056, India; 4Department of Cardiology, Indraprastha APOLLO Hospitals, New Delhi 110076, India; 5Heart and Vascular Institute, Adventist Health St. Helena, St. Helena, CA 94574, USA; 6Department of Radiology, Massachusetts General Hospital, 55 Fruit Street, Boston, MA 02114, USA; 7Department of Electrical and Computer Engineering, Idaho State University, Pocatello, ID 83209, USA; 8Stroke Diagnosis and Monitoring Division, AtheroPoint™, Roseville, CA 95661, USA

**Keywords:** brain tumor, brain tumor characterization, genomics, radiomics, radiogenomics, segmentation, classification, risk-of-bias

## Abstract

**Simple Summary:**

Radiogenomics is a relatively new advancement in the understanding of the biology and behaviour of cancer in response to conventional treatments. One of the most terrible types of cancer, brain cancer, must be targeted in light of the current advancements in therapies. Even though several recent studies on brain cancer have shown promising outcomes when employing the radiogenomics concept, a cutting-edge review of research has been required. In this research review, we provide a 360-degree aspect of brain tumor diagnosis and prognosis employing the new era technology of radiogenomics. The review provides information to the reader about the various aspects that should be considered as accomplishments, opportunities, and limitations in the current therapeutic procedures.

**Abstract:**

Brain tumor characterization (BTC) is the process of knowing the underlying cause of brain tumors and their characteristics through various approaches such as tumor segmentation, classification, detection, and risk analysis. The substantial brain tumor characterization includes the identification of the molecular signature of various useful genomes whose alteration causes the brain tumor. The radiomics approach uses the radiological image for disease characterization by extracting quantitative radiomics features in the artificial intelligence (AI) environment. However, when considering a higher level of disease characteristics such as genetic information and mutation status, the combined study of “radiomics and genomics” has been considered under the umbrella of “radiogenomics”. Furthermore, AI in a radiogenomics’ environment offers benefits/advantages such as the finalized outcome of personalized treatment and individualized medicine. The proposed study summarizes the brain tumor’s characterization in the prospect of an emerging field of research, i.e., radiomics and radiogenomics in an AI environment, with the help of statistical observation and risk-of-bias (RoB) analysis. The PRISMA search approach was used to find 121 relevant studies for the proposed review using IEEE, Google Scholar, PubMed, MDPI, and Scopus. Our findings indicate that both radiomics and radiogenomics have been successfully applied aggressively to several oncology applications with numerous advantages. Furthermore, under the AI paradigm, both the conventional and deep radiomics features have made an impact on the favorable outcomes of the radiogenomics approach of BTC. Furthermore, risk-of-bias (RoB) analysis offers a better understanding of the architectures with stronger benefits of AI by providing the bias involved in them.

## 1. Introduction

A brain tumor is a fatal disease, and its threat to human life makes it more challenging with the escalation in the number of deaths globally [1]. Brain and other nervous system cancer is a prevalent type affecting adult men, women, and children and is the tenth *primary cause* of death [2]. According to the 2018 Global Cancer Registry results, there were nearly 18 million registered cancer cases in both sexes, of which about 30,000 were related to brain cancer [3]. The brain cancer burden continues to rise globally, producing enormous physical, sentimental, and monetary burdens on individuals, families, communities, and healthcare systems. Countries with mediocre and poor healthcare support systems do not have the exposure to on-time standard diagnosis and treatment for many patients [4]. 

The traditional diagnosis of brain tumors includes chemotherapy, radiation therapy, biopsy, and surgery based on the tumor size, type, location, and grade [5]. To provide an add-on to these traditional treatment methodologies, medical imaging techniques have been providing phenomenal results for the treatment. The various medical image modalities that have been used for neurological disorders’ and brain tumors’ detection include magnetic resonance imaging (MRI) [6], computed tomography (CT) [7], cranial ultrasound imaging [8], and positron emission tomography (PET) [9,10,11], of which MRI is considered to be the most desirable option because of its soft tissue characteristics, and radiation-free nature [12]. However, the radiologist’s manual inspection of the medical image to detect tumor features with disease characteristics is tiresome, and the performance varies based on the radiologist’s experience. 

The advancement of artificial intelligence (AI) technologies, such as machine learning (ML) and deep learning (DL), have shown benefits in disease detection and classification [5,13]. Several ML methods have been adapted in cancer such as brain [14], liver [15], thyroid [16,17,18], tumor vascularity in breast cancer [19,20], ovarian [21,22], skin [23,24], diabetes [25,26], heart disease [27], and coronary artery disease diagnosis [28,29,30]. These ML techniques have performed well; however, the feature extraction techniques are ad hoc, causing variabilities in ML results [5]. 

More recently, deep learning (DL) methods came into existence [31,32] and have shown several medical imaging applications such as in brain cancer [33], carotid wall segmentation [34,35], COVID lesion detection, lung segmentation [36,37], and coronary/carotid plaque classification [38,39]. These DL techniques are certainly better than ML, but they are pretty challenging due to the cost of training time. New techniques such as transfer learning (TL) [40,41,42] and hybrid deep learning (HDL) [37,43] have improved the training time for the diagnosis process of brain tumor classification, detection, and segmentation with a greater level of automation. These AI technologies have been able to find the tissue-based characteristics of the disease from the medical image using various feature extraction methods and with the subsequent segmentation and classification of disease characteristics [44]. Hence, the amalgamation of AI with radiography takes the brain cancer diagnosis to the next level.

For early detection, quality treatment, and survivorship, the AI paradigm provides classification, detection, and segmentation for brain tumors and has proven successful so far [14,33]. However, the performance of these classification and segmentation methods needs to be further improved for a better diagnosis rate [45]. Detecting genomics information during brain tumor classification and segmentation can further enhance the diagnosis process as the genetic mutation is the prime cause of brain cancer [46]. Due to the exposure of healthy cells to external substances, there is an alteration in the genes present in the cell; thereby, the cells turn to being cancerous cells [47]. Hence, genetic mutation is the prime cause of brain tumors that should be identified during diagnosis. The genomic assessment and status of genetic mutation on various genes and cell proteins, which are the molecular characteristics of the disease, can be detected from the radiological medical imagery using the AI paradigms [48,49,50,51]. 

In the proposed review for brain tumor characterization using radiomics and genomics features, we start with the biology of brain tumors to understand the underlying process of brain tumors and the mutation process of genomics in the glia brain cells. The essential genetics that have a vital role in brain tumors are discussed with their mutation process and the status of the gene. The classification and grading system of various brain tumors based on genetic mutations have also been tabularized to provide broad information about the brain tumor’s characterization [52]. 

Radiomics and radiogenomics are the two emerging research directions for cancer diagnosis that “finds the disease” and for “genetic” characteristics. Radiomics deals only with the imaging characteristics or phenotype of the disease, while “radiogenomics” links both imaging (phenotype) and genetic (genotype) characteristics of the diseases. The AI technology such as ML, DL, and TL has been helpful in both radiomics and radiogenomics studies in finding out the features of brain tumors accurately and successfully. We thus hypothesize that “AI combined with radiogenomics” can be a powerful paradigm for the diagnosis of brain tumors.

The remaining layout of the article for brain lesion characterization is as follows. Section 2 discusses the PRISMA model search strategy and AI attributes’ statistical distribution in radiogenomics studies. Section 3 presents the biology and mutation process of brain cancer. Additionally, classification and grading information of various brain tumors are presented. Section 4 narrates the essential genetics that cause brain cancer and their summary. The radiomics approach for brain lesion characterization using imaging modality is discussed in Section 5. Section 6 displays the radiogenomics method and some critical discussion on various aspects of brain tumor characterization in Section 7. Finally, the study concludes in the last section.

## 2. PRISMA Model and Search Strategy

This section discusses the radiomics and radiogenomics studies’ search approach using the Preferred Reporting Items for Systematic Reviews and Meta-Analyses (PRISMA) model, statistical distribution, and analysis of several important and impacting AI features.

### 2.1. The PRISMA Model 

The proposed review included the bias analysis in radiogenomics. The PRISMA prototype has been employed for the selection process of relevant publications as shown in Figure 1. The top academic search databases such as PubMed, IEEE Xplore, Google Scholar, MDPI, and Scopus have been used for this purpose. The admissible keywords used for the search are: “brain tumor”, “radiogenomics for brain tumor”, “radiomics”, “genomics”, “radiogenomics”, radiogenomics using AI”, “deep learning for radiomics”, “machine learning for radiomics”, “radiogenomics using deep learning”, and “radiogenomics using machine learning”. A total of 314 records were collected from search databases and other sources. The duplicate records were excluded from this cluster using the “Find Duplicates” feature in the EndNote X9 citation software tool by Clarivate Analytics, which resulted in 212 articles. Again, E1, E2, and E3, as shown in Figure 1, are marked as the three exclusion criteria removing 19, 17, and 11 publications, respectively, under the genre of (i) non-relevant articles, (ii) studies not related to AI, and (iii) articles with insufficient data, resulting in 165 publications. Finally, 20 selected articles [50,53,54,55,56,57,58,59,60,61,62,63,64,65,66,67,68,69,70,71] out of these 165 were identified for the bias analysis of radiogenomics studies for risk-of-bias analysis as discussed in Section 7.3.

### 2.2. Statistical Distributions of AI Attributes for Radiogenomics Studies

#### 2.2.1. Statistical Distribution of Country-Wise Study of Radiogenomics Using AI

With the worldwide evolution of radiogenomics studies using the cutting-edge technology of AI, it is essential to understand the origin and research of different nations. It has been observed that more than 50% of the research on this new technology is conducted in the USA [53,54,59,60,61,65,68,71]. The other major contributor is the Republic of China, with 20% of the total studies conducted throughout the review. However, Japan, South Korea, and Turkey have also been actively involved in this radiogenomics study with contributions of 10%, 5%, and 5%, respectively. Such a nation-wide distribution is shown in Figure 2a. It is observed that other parts of the world are not much involved, which shows some biased nature of the radiogenomics study. It should be expected from the remaining part of the globe to be actively involved in this quite innovative research to take it to a different level.

#### 2.2.2. Statistical Distribution of AI Used in the Radiogenomics’ System

Since the year of inception in 2002 by Andreassen et al. [72], the study of radiogenomics has been developed in many dimensions. The introduction of AI makes the performance of radiogenomics applications better. The AI technologies such as ML, DL, and transfer learning (TL) have been performing the task of automatic radiomics feature extraction and genomics classification with much ease. Due to its recent success, DL technology has been predominately used in computer vision problems, and it covers 50% of the study considered under this review, as depicted in Figure 2b. Under the DL technology, both phenotype and genotype information of the image can be detected automatically [60,61,66,67]. With the advantage of transfer learning, the small dataset on radiogenomics has been able to perform better. In the case of ML, the hand-crafted radiomics features also provide quite better level performances [63,64]. Figure 2c shows the types of AI models for radiogenomics studies. Among the DL-based model, the convolutional neural network (CNN) and ResNet models are the most popular choices among researchers [60,61,65]. However, other DL-based models used are GoogleNet [71], DenseNet [67], VGG [71], and DNN [53,66]. The ML-based models showed a 28% share. The classifier has always played an essential role in an AI classification model. Figure 2d shows the distribution of different types of classifiers. In radiogenomics study for genotype status prediction, various AI-based classifiers successfully used are the softmax type with 38%, random forest (RF) with 19%, and SVM with 13%. A smaller component is taken by the classifiers such as logistic regression, ANN, XGBoost, K-NN, decision tree (DT), and naïve Bayes, which are all less than 10%.

#### 2.2.3. Statistical Distribution by Image Modality and Anatomical Area of Radiogenomics’ System

Imaging modality always plays an important part in the medical imaging system. Figure 3a shows the kinds of medical imaging modalities employed. Magnetic resonance imaging (MRI) and computer tomography (CT) are prominent imaging modalities for all kinds of oncology, with 70% and 30% of studies, respectively. Imaging modality directly relates to the anatomical area considered for imaging. Figure 3b shows the distribution of radiogenomics studies for different kinds of organs. Brain images are primarily compatible with MRI; however, CT and PET are also considered. It can be observed from the proposed review that researchers are more oriented toward the brain parts of oncology study with 45% as compared to other anatomical features of the human body. The remaining important anatomical section is breast cancer, with 25% of the studies. The other critical anatomical sections of the body for oncology study are the lung, pancreas, kidney, and skeletal muscle.

#### 2.2.4. Statistical Distribution of Performance Evaluation on Radiogenomics System

An AI-based diagnosis system’s final and most crucial component is performance evaluation. The better the AI system, the higher the values of the performance evaluation parameters. Accuracy and area under the receiver-operating characteristic curve (AUC) have been chosen as the most appropriate performance evaluation measures in the majority of radiogenomics investigations. However, statistical evaluations such as accuracy, sensitivity, and specificity have been applied to some extent. The mean and standard deviation (SD) of the radiogenomics studies has been found to be 83.83 ± 10.92 and 85 ± 10.30, respectively, as shown in Figure 3c.

#### 2.2.5. Statistical Distribution by Dataset Size Covering all the Objectives and Modalities 

In this review, the dataset size appraised for the radiogenomics research incorporates the number of patients evaluated for the corresponding research. This covers all of the study’s goals and its methodology. The dataset is not easily accessible by the general public and is not volumetric because radiogenomics is a relatively young area of study. The datasets used in various studies under this review are basically systematic electronic radiological data with open access such as The Cancer Imaging Archives (TCIA) [60,61], The Cancer Genome Atlas (TCGA) [73], The Quantitative Imaging Networks (QIN) [74], and brain tumor segmentation (BraTS) [67], and some have been collected from various hospitals and consortiums [53,63,64,66]. Moreover, we further analyze some other important brain tumor cohorts with a large collection of genotype features. The Repository of Molecular Brain Neoplasia Data, termed *REMBRANDT* [75,76,77], provides a facility for establishing a connection between genetic data and clinical information. It has data of 874 patients suffering from a glioma containing gene expression arrays. Furthermore, it contains MR scans of 130 REMBRANDT patients. Overall, this dataset offers an opportunity to explore brain tumors in current clinical practice in the era of precision medicine. Moreover, *VASARI* (Visually Accessible REMBRANDT Images) [78,79] was used for defining quantitative features from the MR scans of brain tumor (glioma) patients using the REMBRANDT project. This set of features has 24 observations for neuroradiologists that describes the brain tumor.

Dealing with all kinds of medical research data, including cancer, has been an important issue for both researchers and the healthcare industry. To develop a robust AI system for cancer research, open access to resources such as gene expression and molecular characteristics and radiomics data must be available to the researchers [80,81]. Genomic and molecular information obtained through multiple tissue samples provides detailed spatial information of tumors at different locations [82]. This information could be used for the heterogeneity analysis of tumors. The research band in cancer is narrowing because of the limited availability of open and public databases. To access important cancer research data available under various super specialist cancer hospitals has become even more restrictive. There are several reasons for such restrictions, such as many patients not wanting to participate, and sometimes hospitals show their restrictive behavior, patients leave their follow-ups, etc. To produce an efficient AI model for brain cancer studies, a large cohort with follow-up information is highly recommended. In our opinion, there could be a solution for such restrictions and an increase in the cohort size in the future by collaborating with multiple institutes, in which institutes will have to share the trained AI model (based on certain standard guidelines) instead of sharing the patient’s information. Furthermore, based on these trained models, further analysis (test) will be conducted. Moreover, the restrictions imposed will be scarcer due to the incorporation of privacy-preserving methods. Even more controlled-access initiatives have been taken to protect the privacy of research participants [83,84]. It is noticed that all the studies incorporated have datasets within 1000 subjects, and a few studies [50,54,59] have been limited to below 100 subjects, as depicted in Figure 4. To minimize data imbalance and overfitting circumstances, a larger amount of data is anticipated for a more accurate assessment of the radiogenomics’ AI system.

## 3. Biology of Brain Tumor 

The biology or pathophysiology of brain tumors is discussed in this section. This includes the cells, the brain cell (glia and neuron), deoxyribonucleic acid (DNA), mutation, mutation process, and uncontrol growth of the mutated cell.

### 3.1. Brain Glia Cell and Deoxyribonucleic Acid 

The cell is the smallest unit of an organism with a nucleus and the molecular fragments of the human anatomy that determine the purpose of each organ within the body. The main job of the cell includes blood flow, oxygen flow, and metabolic management in the human body. However, the most regular brain cells are neurons and non-neuron cells (called *glia* cells) [85]. These neurons have an essential role in the proper functioning of the brain and passing signals to other parts of the body. The glia cells are the supporting cast of the nervous system that fuel the function of the neurons. The neuroglia cells are the non-neural cells in the central nervous system (CNS) and peripheral nervous system (PNS) that do not produce neurological electrical signals [86]. The average adult human brain has about 100 billion neurons, with glia cells accounting for ten times that number.

There are six types of glia cells having different functions, such as (a) astrocytes, (b) oligodendrocytes, (c) microglia, (d) ependymal cells, (e) satellite cells, and (f) Schwann cells. The first four glia cells belong to the CNS and the last two to the PNS, as shown in Figure 5b. *Astrocytes* are star-shaped cells that keep a neuron’s operating environment in a good condition. They provide protection and structural support to neurons, exchanging nutrients and other important chemicals with neurons. *Oligodendrocytes* help neurons in the central nervous system, especially those traveling vast distances within the brain, by supporting their axons. They make a fatty material called myelin, which forms a protective covering surrounding the axon. *Microglia* scavenge and destroy dead cells while protecting the brain from microbes. *Ependymal cells* line the spinal cord and ventricles of the brain. They help make cerebrospinal fluid (CSF), which serves as a brain cushion and transfers fluid between the spinal cord and the brain. It is also a component of the choroid plexus. The *Satellite glia* [87] supplies nutrition and structural support for neurons in the PNS. *Schwann cells* of the PNS are similar to oligodendrocytes in the CNS, which myelinate neurons in the PNS.

The nucleus, which contains 23 pairs of chromosomes and millions of genes, is the core control system of each glia cell. [33]. Deoxyribonucleic acid (DNA), which acts as a blueprint for genes and specifies their function, contains the instructions for these genes.

### 3.2. Mutation and Its Process

A mutation causes a change in the DNA sequence, which is the fundamental cause of gene malfunction. DNA mutations are influenced by a variety of factors, including the environment, lifestyle, and eating habits. Three types of cancer-causing genes have been identified such as (a) tumor suppressors, which control the cell death cycle (apoptosis), (b) genes that are accountable for the restoration of the DNA, and (c) proto-oncogenes, which work in opposition to tumor suppressor genes and are responsible for the synthesis of a protein that promotes cell proliferation while blocking standard cell death [33].

### 3.3. Central Nervous System and Brain Tumor Types’ Classification and Grading

The knowledge of classification of brain tumor types and grading provides many advantages in treating the various types of tumors. The World Health Organization (WHO), the apex organization of world health, has defined the latest, i.e., the fifth issue of classification of tumor types in 2021 following the initial publications from 1979, 1993, 2000, 2007, and 2016. The new edition of tumor types’ classification basically focuses on molecular information with histology that helps more accurate diagnoses, prognoses, and treatment plans, and to predict therapeutic responses for patients. The recent advances relate to classification approaches according to the WHO: more than 120 brain and central nervous system (CNS) tumors exist for both adult and pediatric types. The various types and subtypes of CNS and brain tumor classification systems have been illustrated by Louis et al. [52] in Table A1 in Appendix A. Apart from the *classification system* of brain tumors, the WHO also provided the *grading system* of CNS and brain tumors based on the cancer type’s severity. Table 1 lists the four grading systems and some selected brain tumor types associated with each grade.

## 4. Genetics of Brain Tumor 

### 4.1. Genomics Types’ Information of Brain Tumor 

The genes and enzymes of the human body play an essential role in human growth and nourishment. Due to the various external parameters based on alteration, these genomes and enzymes may cause brain cancer [89]. Here, we will discuss prominent genotypes that are accountable for the formation of brain tumors and discuss their impact and severity, such as glioblastoma multiforme.

Isocitrate Dehydrogenase (IDH): IDH1 and IDH2 are two families of a gene that generate enzymes called isocitrate dehydrogenase (IDH), which are helpful for the breakdown of nutrients and produce energy for cells [33,89,90]. The enzymes produced from these genes are involved in the amino acid and citric acid formation cycle. The mutation of these genes is strongly associated with glioma and other types of cancer. Figure 6 shows the chemical reaction and metabolic pathways present in a brain tumor cell, with emphasis on enzymatic effectors—IDH1 and IDH2 mutations. IDH1 gene mutation is found in 5% of patients having a primary glioblastoma; however, 70% to 80% of patients develop a glioma of grade II to III and secondary glioblastoma [90]. IDH2 mutations are primarily seen in oligodendroglia tumors. Another IDH variant is the IDH-wildtype gene, which has a more rigorous form of IDH mutation, with distinctive genetic and clinical characteristics than IDH1 and IDH2. Patients with the IDH-wildtype gene present have 90% chance of a glioblastoma developing and carry a worse prognosis than other IDH mutants. 

The systematic review and meta-analysis conducted by Suh et al. [91] for IDH (isocitrate dehydrogenase) mutation prediction in glioma patients with the help of imaging radiomics data summarizes: Compared to IDH-wildtype glioma, IDH-mutant glioma consistently displayed less aggressive imaging characteristics. MRI demonstrated the potential to non-invasively predict IDH mutation in patients with glioma despite the wide range of different MRI techniques used. The pooled sensitivity of 2-hydroxyglutarate MRS is higher than that of other imaging modalities.

Yan et al. [89] presented the IDH1 and IDH2 mutation in glioma, which is the most frequently occurring brain tumor in the adult brain and central nervous system. The conclusion of the study reveals that in the majority of the various forms of malignant gliomas, mutations of the NADP + dependent isocitrate dehydrogenases encoded by IDH1 and IDH2 are found as shown in Figure 6. 

1p and 19q Co-deletion: The mutation of 1p and 19q co-deletion implies the compound loss of both the short arm of chromosome 1 (1p) and the long arm of chromosome 19 (19q) is the molecular genetic signature of oligodendrogliomas, a subtype of primary brain tumors accounting for approximately ten to fifteen percent of all diffuse gliomas in adults [93]. The 1p and 19q co-deletion chromosomes help in the glioma’s prognosis, diagnostic, and predictive biomarker. Hence the mutation effect of the gene results in a poor prognosis of the brain tumor [33]. Up to 80% of oligodendrogliomas, 60% of anaplastic oligodendrogliomas, 30–50% of oligoastrocytomas, and 20–30% of anaplastic oligoastrocytomas have the mutation impact of this gene [90].

A systematic review and meta-analysis by Hu et al. [94] for the role of chromosomal 1p/19q co-deletion in connection with personalized treatments for patients suffering from oligodendrogliomas demonstrate that the pooled findings of the study show the prognosis of grade II/III oligodendrogliomas is significantly improved by chromosomal 1p/19q co-deletion.

The research study conducted by Han et al. [95] for the development and validation of an MRI-based radiomics signature for the non-invasive genotype prediction of chromosome 1p/19q co-deletion in lower-grade gliomas summarizes that an MRI-based radiomics signature may successfully detect the 1p/19q co-deletion in lower-grade gliomas (WHO grade II and III gliomas) with histological diagnosis, potentially enabling the non-invasive genetic subtype prediction of gliomas.

O6-methylguanine DNA methyltransferase (MGMT): The MGMT is a DNA rebuilding enzyme that shields the tumor cells from the alkylating agent (induced damage during chemotherapy) by eliminating alkyl groups from the O^6^ position of the guanine [82]. The MGMT promoter methylation has been established as an important clinical biomarker in neuro-oncology. The presence of the promoter has a better prognosis and longer survival with glioblastomas who receive alkylating agents. It is useful for identifying newly diagnosed glioblastomas and also helps in a patient’s survival prediction [33,90]. MGMT promoter methylation has also been reported to have high rates in oligodendrogliomas and astrocytomas of lower grade, which variably correlate with 1p19q co-deletion and IDH mutations. Methylation of the MGMT promoter is detected in 35–75 percent of glioblastomas. [96].

The review research by Cabrini et al. [97] on the most common genomics biomarkers of brain cancer, i.e., MGMT for the regulation of expression of MGMT and treatment of glioblastoma, reviews various aspects such as the role of MGMT in cancer, nuclear transcription factors that control MGMT gene expression, the effect of histone modifications, the MGMT promoter, regulation of MGMT expression by microRNAs, MGMT as a prognostic biomarker, and, finally, the manipulation of MGMT expression to enhance first-line glioblastoma treatment.

Silber et al. [98] in their review of the promise and problems of O6-methylguanine DNA methyltransferase in glioma therapy, suggest the improvement in the utility of MGMT methylation status in planning optimal therapies tailored to individual patients.

B-Raf proto-oncogene (BRAF): The BRAF is a proto-oncogene that encodes the protein B-Raf of the RAF family of serine/threonine protein kinases. This protein influences cell division, differentiation, and secretion via modulation of the MAP kinase/ERK signaling pathway [99]. This gene’s mutations, most notably the V600E mutation, are the most commonly detected cancer-causing mutations in melanoma and have also been found in various other malignancies. [100].

Werner et al. [101] in their review research on isocitrate dehydrogenase and B-Raf proto-oncogene-wildtype epithelioid glioblastoma treatment response viable on MRI radiological imaging features, suggest that this subtype of genomic biomarkers is frequently linked to a poor clinical outcome and overall survival. Moreover, epithelioid glioblastoma without an isocitrate dehydrogenase or B-Raf proto-oncogene mutation was found to be a very aggressive molecular subtype by neuropathology.

The systematic review by González-González et al. [99] on B-Raf proto-oncogene serine and threonine kinase V600E mutation for ameloblastoma-targeted patients’ treatments summarizes that when surgical treatments for ameloblastomas are not an option, the discovery of these genomic biomarkers has revolutionized alternative therapies.

Tumor Protein 53 (TP53 or p53): The TP53 gene has a vital role in the formation of a protein called tumor protein (p53) and also in DNA repair [33]. The TP53 also acts as a tumor suppressor, a transcription factor that controls the cell proliferation from an uncontrolled growing and dividing of cells and regulates the cell cycle. The alteration and abnormalities of TP53 lead to genetic instability, reduced apoptosis, and angiogenesis [90]. The degree of mutation is more relevant to high-grade glioma and 80% of the tumor. It is otherwise called the guardian of the genome as it repairs or self-destructs (apoptosis) the DNA when there is damage by external agents such as radiation or ultraviolet (UV) rays from sunlight and toxic chemicals. 

Zhang et al. [102] in their review research on “The p53 Pathway in Glioblastoma” summarize the following statements. One of the most often dysregulated genes in cancer is TP53. In 84 percent of glioblastoma (GBM) patients and 94 percent of GBM cell lines, the p53-ARF-MDM2 pathway is dysregulated. Components of the p53 pathway that are dysregulated have been linked to GBM cell invasion, migration, proliferation, apoptosis evasion, and cancer cell stemness. Numerous microRNAs and long non-coding RNAs are also involved in the regulation of these pathway constituents. The majority of TP53 mutations in GBM are point mutations that increase the expression of the p53 protein’s gain of function (GOF) oncogenic variants. These GOF p53 mutants, which have received relatively little research, may cause GBM malignancy by acting as transcription factors on a different set of genes from those controlled by wildtype p53. Their expression is associated with a worse outcome, suggesting that they may be important indicators and treatment targets for GBM.

Retinoblastoma1 (RB1): The RB1 gene helps in the formation of protein in a cell called P^RB^. This RB1-generated protein works as a tumor suppressor that regulates the growth and uncontrolled division of cells. The mutation of this genomics hence blocks cell cycle progression and unchecked cell cycle progression. The tumor suppressor pathway is commonly seen in the case of high-grade gliomas such as glioblastoma multiforme (GBM). Additionally, the RB1-generated protein influences cell survival and cell self-destruction (apoptosis) [33,90].

Goldhoff et al. [103] in their proposed research, focus on the clinical classification of glioblastoma based on retinoblastoma tumor suppressor protein (RB1) mutations and link with the proneural subtype. The result analysis suggests that clinical GBM specimens using IHC to determine RB1 status, and they hypothesize that RB1 changes may be more prevalent in specific GBM subtypes.

Epidermal Growth Factor Receptor (EGFR): The EGFR is a transmembrane protein that is a receptor for members of the epidermal growth factor family such as phospha tase and tensin homolog (PTEN), receptor tyrosine kinase (RTK), and phosphatidylinositol 3-ki nase (PI3K) cell proliferation pathway. The mutation of this protein increases the cell proliferation rate. Upregulation or amplification of the EGFR gene has been linked to various malignancies, including 50 percent of primary glioblastoma cases [33,90].

The review research on “Epidermal Growth Factor Receptor (EGFR) in Glioblastoma” by Xu et al. [104] focuses on various aspects such as EGFR mutation in glioblastoma, EGFR therapies in glioblastoma, and glioblastoma’s dysregulated EGFR signaling networks such as PI3K, MAPK. 

Phosphatase and Tensin Homolog (PTEN): Phosphatase and tensin homolog (PTEN) is a phosphatase in humans that is encoded by the PTEN gene. Mutations in this gene cause a variety of malignancies, including glioblastoma, lung cancer, breast cancer, and prostate cancer. It is also a tumor suppressor gene. The mutation effect of this gene has increased cell proliferation and reduced cell death in 80% of glioblastomas [33,90].

Karsy et al. [105] in the practical review on prognostic correlations of various molecular biomarkers in glioblastoma including the phosphatase and tensin homolog (PTEN) discuss the clinical trials on these biomarker indicators and their applicability to clinical practice, the distinction between prognostic and predictive biomarkers, and other related information. Furthermore, the survey finds that despite PTEN’s significance in the development of gliomas, there is still no evidence linking it to survival.

### 4.2. Brain Tumor Types and Their Associated Genes

Brain tumors are associated with several familial cancer predisposition syndromes. This portion of the article depicts the key genomes and proteins whose germline mutation forms the different types of brain tumors, as provided in Table 2.

## 5. A Radiomics Approach to Tumor Characterization

This section is based on radiological imaging or radiomics to characterize the brain tumor with segmentation and classification as part of the radiomics process. To proceed into the central paradigm of brain tumor characterization, we enlighten the understanding of the pre-requisite concepts such as imaging modalities and the functionality of the modality.

Magnetic resonance imaging (MRI) is a non-invasive, highly sophisticated, highly detailed, three-dimensional imaging diagnostic method of the interior structures of the human body [14,109]. In neurology and neurosurgery, MRI is one of the most often used tests. MRI has an advantage over CT in detecting flowing blood and cryptic vascular malformations. It can also detect demyelinating disease and has no beam-hardening artifacts such as the damaging ionizing radiation of x-rays that can be seen with CT. Due to the non-use of ionizing radiation in X-rays, this MRI imaging modality is used when there is a need for frequent imaging for diagnosis and therapy of the brain. The brain, spinal cord, and nerves are more easily visualized on MRI than on CT [110]. The MRI is the preferred imaging diagnostic method for head and neck injuries, cancers, and hemorrhages. Table 3 provides the comparison of prominent imaging modalities based on various factors.

The MRI has several limitations [90,111]. A magnetic resonance imaging (MRI) scan is a painless and safe procedure that gives crisper images of the body and its tissues from any angle. This is very helpful for spotting soft tissue cancers all over the body. While an MRI scan is a reasonably safe process that does not expose you to harmful radiation, some dangers are still associated. However, MRI safety has recently become a significant emphasis in hospital and outpatient settings because of the possible attraction to ferromagnetic materials and devices. In comparison to conventional X-ray and CT scan procedures, an MRI is an exceedingly high-cost and time-consuming inquiry. The bone, bone injury, and acute brain hemorrhage are better examined using CT and X-ray techniques. It also generates motion artifacts. Some illness processes may be challenging to distinguish with an MRI. Because of the longer time it consumes and all other instruments must be vacated from the room while the MRI device runs, it is not the right investigation for emergencies. This restricts its application in trauma and emergencies, where CT scanning is frequently favored. The other details regarding the MRI and other imaging modalities used in brain tumor analysis have been provided in Appendix B.

### 5.1. Radiomics of Brain Tumor Characterization Using AI Framework

Radiological scans have proven to be an effective non-invasive tool in the screening and diagnosing of early-stage brain cancer, treatment strategy assistance, prognosis evaluation, and follow-up for advanced-stage brain cancer [112]. Recently, radiological features have evolved from semantic to radiomic hand-crafted and deep features [48,113]. Semantic features are the qualitative features commonly extracted by the experienced radiologist directly by analyzing and looking at the clinical image to describe the lesion [114,115]. However, radiomic features entail extracting and analyzing quantitative features from medical images using mathematical algorithms, machine learning, and deep learning methods to explore possible ties with biology and clinical outcomes [116]. The anatomical and functional information of the lesions can be better viewed by the radiomic features extracted from structural and functional images [117]. *Structural imaging* is a term that describes ways of visualizing and analyzing anatomical data from the human body, such as the brain structure, tumor location, traumas, and other brain problems. *Functional imaging* is a term that describes methods for studying tumor physiology and molecular processes such as metabolic alterations, finer-scale lesions, and viewing brain functions [113,118,119].

#### 5.1.1. Traditional Radiomics

In the traditional radiomics approach, the hand-crafted features of radiological images play an essential role. Using the AI framework, the conventional radiomics workflow involves many steps such as image acquisition, pre-processing, region of interest segmentation, feature extraction, feature reduction, model selection, evaluation, and validation in clinical implementation [120]. The vital steps of all the workflows are the radiomics feature extraction and selection from the delineated region of interest. The hand-crafted radiomics features have been generated by software implementing mathematical algorithms. The input image of manifold modalities such as MRI, CT, PET, etc., with the region of interest, is provided to a mathematical model to obtain the feature values [121,122]. The mathematical formulations of the features generated are independent of the imaging modality [123]. 

The wide range of hand-crafted radiomics features derived from medical images includes intensity, texture, geometric, dynamics curve, morphological, metabolic, angiogenesis, spatial, histogram, and statistical features [42,115]. These features’ traits, known as radiomic features, may reveal tumoral patterns and characteristics that are not visible to the human eye. The texture and intensity feature provides the pixel and voxel label representation of a tumor and their corresponding intensity and texture representation. The radiomics texture features can serve as potential biomarkers for determining *BRAF* mutation status and as predictors [124]. The radiomics geometrics feature describes the tumor’s shape, size, volume, diameter, surface, and compactness. The statistical features of the clinical images such as the mean, median, mode, maximum, minimum, variance, standard deviation, percentiles, energy, entropy, skewness, uniformity, kurtosis, bias, histogram, and dynamics curve demonstrate the statistical distribution of pixel or voxel intensities of the tumor region. 

These radiomics features have provided inputs to the various supervised, semi-supervised, and unsupervised machine learning models such as support vector machine (SVM), K-nearest neighbor (K-NN), decision tree (DT), random forest (RF), naïve Bayes, ensemble, artificial neural network (ANN), logistic regression, clustering, and auto-encoder for classification of a brain tumor’s characterization. Finally, data analysis with the outcome of radiomics includes survival prediction, personalized treatment planning, and precision medicine.

#### 5.1.2. Deep Radiomics 

The deep radiomics study belongs to the DL technology, a subset of ML. The well-known DL models such as convolutional neural network (CNN) and recurrent neural network (RNN) have been successfully used in computer vision studies. The pre-trained model of CNN, such as AlexNet, ResNet, GoogleNet, etc., training on the voluminous ImageNet dataset, showed quite good performance of image features of the multi-set datasets [125]. The transfer learning (TL) also adds dimensionality to the deep feature extraction and successful computation with a suitable classifier. However, the deep radiomics study with deep feature extraction and subsequent classification is marginally better than the conventional hand-crafted radiomics study. The automatic feature extraction methods under deep radiomics select the necessary and proper features from the radiological medical image. The deeper the deep learning network, the more powerful features are extracted, and they are more abstract and meaningful and passed to the next layer of the network. The single deep learning model can perform radiomics feature extraction, selection, and final data analysis with the classification of brain tumors. In contrast, multiple models are involved in the case of traditional radiomics.

Image pre-processing is essential when dealing with clinical images, both for traditional and deep radiomics. Due to the varying intensity of the medical image, the presence of salt, noise, and blur, and the position of the scanners, this step is of utmost importance and enhances the performance of the model. The various pre-processing processes involved in brain tumor imaging are denoising [126], normalization, bias correction (N4ITK), image registration [29,127,128], skull stripping, and histogram matching. After the pre-processing, the region of interest (ROI) is detected, from where radiomics features are extracted. As discussed earlier, the radiomics features may be hand-crafted or deep. The final step of the radiomics process is the model selection, and data analysis from the radiomics features for better clinical decision making and treatment planning, as shown in Figure 7. 

## 6. AI Modelling Buffered with Genetics for BTC: A Radiogenomics Approach

Brain tumor classification and segmentation have been the most desired choices of the radiologist for the diagnosis process and the detection of the brain lesion. The intervention of AI for the process of brain tumor segmentation and classification has made automation in the diagnosis process and helped the radiologist to a greater extent for the detection of the lesion and to classify the tumor accurately. As discussed, genomes and genetic mutation have been the primary causes of brain tumors. Hence, detecting the status of various genomes in genetic mutation in the brain tumor can further help the diagnosis and prognosis process in a personalized treatment plan. Using ML and DL technologies, AI can detect the genetic status from radiography [129,130].

Table 4 summarizes some helpful research articles and their findings that use AI, radiomics, genomics features, and performance. The MRI is the most suitable scan for any brain disease and brain cancer because of the soft tissue level imaging. The multi-parametric (T1, T1CE, T2, FLAIR, SWI, DWI, etc.) nature of MRI further improves and analyses the cancer tissues and provides their exact shape, size, location, and texture. The contrast is enhanced with external agents for MRI modalities such as T1-CE, SWI, and DWI and is more able to predict the genomic status from the better cancer features available in them [53,63,64,131]. From the observation, deep radiomics is the preferred radiomics feature over conventional hand-crafted radiomics features because of its deep, automatic feature extraction, and selection process. This further indicates the use of various DL models over ML models to classify and detect genomics status with better performance [53,60,61,66,67,68,131]. Among the various deep model signatures, the ResNet and its variants models have been considered the most desired because of the residual block to handle vanishing gradient features [60,61,65,67]. Out of all molecular signaling responsible for brain tumors, IDH and MGMT are the most common genes whose mutation causes brain cancer; hence, the prediction of the mutation status of these genes is more important for better prognosis [60,61,63,131]. The accuracy, which is the very commonly used performance metric of AI models and AUC is preferable for classification problems with probabilities to analyze the prediction more profoundly and has been used here for genetic status prediction.

### 6.1. Radiogenomics Pipeline for Brain Tumor Classification Using AI Model 

The radiogenomics workflow for brain tumor genomics and disease characteristics can be divided into five steps, as shown in Figure 8. 

Step 1. Data acquisition and pre-processing: At this stage, both radiomics and genomics data have been acquired and fed into the AI model for further processing. For the brain radiomics data, MRI scans of various modalities are preferred, and these images need pre-processing. It is wise to pre-process medical image modalities because of the various artifacts present. Additionally, medical images are captured under various scanners and consider the situation with the severity of patients. Intensity normalization, skull stripping, bias correction, denoising, and histogram matching are the major steps for pre-processing brain tumor medical images [132]. However, data augmentation is also essential while training under DL- and ML-based AI models [41,42,133,134]. Another necessary procedure is finding the region of interest (ROI) to focus on the disease characteristics and genomics features. For acquiring genomics data, a biopsy has been considered a suitable method to obtain the genomics information on the brain tumor [51]. 

Step 2. Feature extraction and selection: The radiomics features may be hand-crafted, leading to a conventional radiomics approach with subsequent ML models for lesion detection and classification. They may be deep radiomics features, where a deep learning model extracts deep, automatic, and desirable features, and the same DL model is used for subsequent segmentation, classification, and further analysis. Similarly, under the study of radiogenomics, obtaining various genomics data or genotypes features such as MGMT, IDH, EGFR, TP53, HER2, Ki-67, PTEN, etc., corresponding to the radiomics features of the patient is needed for the mapping of phenotype and genotype information and then the subsequent analysis [51,135]. 

Step 3. Association of radiomics and genomics: The genotype–phenotype mapping is a conceptual model in genetic architecture in the context of evolutionary biology and was first coined by P. Alberch in 1991 [136]. This amalgamates radiomics features (phenotypes) with the corresponding genomics features (genotypes) to generate the radiogenomics profile for a particular patient. In this mapping, the biological information is stored and passed on in the form of genotypes and expressed as phenotypes. The structural characteristics of mapping show that there may be phenotypic redundancy, where many genotypes map to the same phenotypes, a highly non-uniform distribution of the number of genotypes per phenotype, high phenotypic robustness, and the capacity to reach a large number of novel phenotypes in a limited number of mutational steps [137,138]. Finally, the genotype and phenotype information mapping are then passed to the AI model for the final analysis.

Step 4. Data analysis: The resulting stage of the whole radiogenomics process is the data analysis stage with appropriate model selection for genomics status prediction from the radiomics features. The classifier at this stage may be a supervised ML-based classifier such as a support vector machine (SVM), random forest (RF), K-nearest neighbor (K-NN), decision tree (DT), naïve Bayes, artificial neural network (ANN), and ensemble classifier. The various DL models involved at this stage use the softmax classifier. Apart from these, statistical models are used for statistical analysis, and performance analysis metrics such as confusion matrix and area under the ROC curve (AUC) are also used for the quantitative measurement of performance. 

Step 5. Outcome: Radiogenomics holds the promise of improving diagnostic accuracy, prognosis assessment, and treatment response prediction, giving useful information for patient care throughout the course of the disease, given that this information is easily obtainable in imaging [139]. As shown in the workflow model, the outcome of radiogenomics involves survival prediction, tumor grading, clinical decision making, precision medicine, and personalized treatment.

### 6.2. Challenges of Radiogenomics

Recently, radiogenomics has evolved as a translational field of research with new trends in the diagnosis and prognosis of cancers and also being applicable to other severe diseases. It provides the best understanding of cancer’s biology and behavior in response to standard therapy. Radiogenomics is the combination of “radiomics” and “genomics,” which has significantly drawn the attention of researchers to determine imaging surrogates for genomic or molecular signatures and to advance biomarkers leveraging the numerous data types used to characterize cancer. However, the new era of technology has been confronting many issues so far from its development.

In comparison to conventional brain imaging, radiomics offers meaningful, high-dimensional, quantitative data of clinical images connected to useful biologic properties. However, the acquisition, availability, accessing, and pre-processing of these radiological imaging databases is an overhead. Firstly, this kind of imaging requires a high-end scanner for its data acquisition, which is more expensive when compared to the standard scanner. Secondly, the open and public availability of these radiomics imaging databases is very limited because they are sophisticated and costlier to deal with. Thirdly, as a consequence of their availability, accessing this imaging is again more complicated due to several restrictions. To develop a robust radiomics imaging database system for cancer research, open access to resources such as gene expression and molecular characteristics and radiomics data must be available to the researchers [80,81]. Genomic and molecular information obtained through multiple tissue samples provides detailed spatial information of tumors at different locations [82]. This information could be used for heterogeneity analysis of the tumor. The research band in cancer is narrowing because of the limited availability of open and public databases. To access important cancer research data available under various super specialist cancer hospitals is even more restrictive. There are several reasons for such restrictions, such as many patients who do not want to participate, and sometimes hospitals show their restrictive behavior, patients leave their follow-ups, etc. Finally, the pre-processing of radiomics datasets is mandatory and requires high-end complex algorithmics. The pre-processing task should be so sophisticated, as it preserves the molecular information of the disease and there should be no leakage of genomics or molecular information. Again, as these radiological images are collected from different scanners of different resolutions and specifications, intensity normalization is the minimum requirement as a pre-processing step. Ultimately, the radiogenomics approach for cancer studies requires a mandatory delicate pre-processing job for suitable feature extraction and selection and subsequently better performance accomplishment. 

The other challenges include the region of interest (ROI) generation, challenging model selection, critical analyzing tools, and, finally, the clinical validations of the radiogenomics outcomes. After the radiomics pre-processing steps, accurate region of interest (ROI) and segmentation of the lesion regions is the most difficult phase of radiogenomics that has a significant impact on the outcome and robustness of the system. Again, due to tumors’ heterogeneity in the shape, size, location, texture, and polymorphic nature of cancer, manual ROI generation has been a tedious type of task. Hence, for the volumetric radiological data, deep learning-based segmentation methods such as U-net [140], U-Net++, V-net [141], etc., have been the first choice for ROI generation. Due to large quantitative radiomics features being present in the radiological images, in most cases, the deep learning-based imaging signatures automatically extract and select the desired and meaningful deep features rather than hand-crafted traditional machine learning algorithms for conventional radiomics features [142,143,144,145]. Again, to handle the volumetric features and outcome results generated by the top-selected model, heavy computational and data visualization models are required for the data analysis and statistical pattern generation. Hence, mostly, the deep learning model, hybrid deep leaning model, machine learning algorithmic models, statistical tools, heatmaps, and grad-cam models are necessary for this step. Finally, the clinical validations of the radiogenomics outcomes state that the current standards, particularly for retrospective data, lack outcomes validation, have insufficient reports of results, and have unidentified confounding variables in the source database.

## 7. Critical Discussion

### 7.1. Principal Findings

Brain tumor characterization is an essential aspect of the diagnosis and prognosis of brain tumor disease. It helps in analyzing underlying disease characteristics, such as the genetic mutation status due to the type of brain tumor that occurs. The proposed review emphasizes the genetic behavior of the brain tumor. For that purpose, we present a comprehensive list of the various kinds of brain and CNS tumors and map the key genomics associated with the kinds of tumors in Table 2. The disease characteristics of cancer can be best analyzed from the radiology features (phenotypes or radiomics) and from the genetic features or data (genotypes or genomics). Hence, the emerging technologies for cancer study are radiomics and radiogenomics approaches. In the case of the radiomics approach, only the radiological image is considered, from which the radiomics features of diseases are extracted and analyzed. However, under the study of radiogenomics, both radiomics and genomics features of the disease are considered to analyze the status of the mutation in brain tumors. Radiomics and radiogenomics, the emerging fields, deal with all kinds of cancer diagnoses and prognosis processes for personalized treatment. However, our statistical analysis suggests extensive research has been conducted on brain tumors (Figure 3b). Therefore, magnetic resonance imaging (MRI) is a suitable imaging modality for brain scanning as it provides soft tissue level characteristics in images without any harmless ionizing radiation, unlike in the case of CT and X-ray imaging (Figure 3a).

The AI has performed supremely in brain cancer detection and diagnosis in the recent past. Hence, it makes the use of radiogenomics approaches for brain tumor detection, diagnosis, and even for personalized treatment and medicine more successful. Out of the ML, DL, and TL models of AI, the DL and DL with TL models are the most suitable and adaptable for radiogenomics as they deal with automatic deep radiomics features extraction and selection for the analysis of brain tumor mutation status (Figure 2b). The various successful DL models involved in the radiogenomics study for brain tumors are convolutional neural network (CNN), VGG, ResNet, DenseNet, and GoogleNet (Figure 2c). This signifies the use of a deep learning-based softmax classifier for predicting the status of genomics mutation in brain tumors (Figure 2d). Therefore, it proves that AI paradigms are advantageous in brain tumor diagnosis and prognosis with radiogenomics. Recently, emerging technologies have helped to a great extent in cancer diagnosis research, with most of the research carried out in the United States of America (Figure 2a).

### 7.2. Benchmarking 

Tumor characterization by the emerging technology of radiomics and radiogenomics has been an active research area among researchers in the recent past. Very little research has been conducted using the AI framework to predict genetic mutation status in various cancer. Additionally, there are a few reviews regarding tumor characterization in the radiomics and radiogenomics paradigms that have been published [49,51,113,114,135,139,146,147,148]. However, only a few are dedicated to brain tumors [49,135,148]. Even though these three studies [49,135,148] dedicatedly discuss the genetic characteristics of brain tumors, they have not covered several aspects for review altogether. Overall, the review articles on cancer characterization are summarized in Table 5. Some did not discuss the AI framework on the prospect of radiomics and radiogenomics. However, some of them discussed only radiomics and only radiogenomics parts separately and may miss the AI framework. Hence, they do not consider an all-round review of these emerging technologies. The proposed review on specialized cancer (brain) characterization, while considering radiomics and radiogenomics individually in the AI framework, our specialization also includes the statistical portrayal of radiogenomics studies of cancer characterization along with risk-of-bias (RoB) analysis. 

### 7.3. Risk-of-Bias Analysis of Radiogenomics Studies

We have considered 20 radiogenomics studies using AI such as DL, ML, and TL to check the bias in these studies to show that AI is good for radiogenomics studies for oncology care. Each study has 19 AI-based attributes such as phenotype, genotype, AI model used number of AI model used, anatomical area of the tumor, image modality, sub-image modality, image pre-processing, dataset size, demographic information of dataset, feature extraction type, feature selection type, data augmentation, classifier information, statistical analysis performed, number of performance evaluation parameters, accuracy, area under ROC curve, and impact factor of the journal. These radiogenomics attributes using AI features are initially qualitative and then quantified by assigning a number between 1 and 5 based on the consensus of the AI scientist’s experience. The value of AI-based attributes has been assigned based on the strength of the attribute that ranges from 1 to 5 of five different classes such as low, moderate, high-moderate, low-of-a-high, and high-of-a-high. Then each individual study’s aggregate score is the sum of all attribute values for that selected study. The mean of each study was then calculated by dividing by the number of AI attributes considered (i.e., 19 in our case). Using this principle, all the studies (i.e., 20) are ranked based on their mean score, which ranges from 4.26 to 3.33, and plotted in decreasing order as shown in Figure 9. The raw cut-off of 3.57 was determined based on the intersection of the “cumulative plot of the mean score” and “the frequency plot curve of the studies”. This raw cut-off mark estimates the whole number of studies into low-bias and high-bias categories. The higher the mean value, the lower is the risk-of-bias; hence, studies above the cut-off mark belong to the low-bias category, while 35% of studies [53,54,55,56,57,58,59] belong to a high-bias category. The highly biased studies have not considered all AI attributes while evaluating the radiogenomics’ system and may have the lower proportioned values for the attributes considered. 

### 7.4. Advanced Features of Artificial Intelligence

There have been two significant developments in the field of AI that cannot be ignored, namely, (a) pruning of AI (PAI) models and (b) explainable AI (XAI). The concept of pruning AI is motivated by the amount of storage needed during the training process of the AI models [149]. In contrast, the concept of explainable AI is inspired by the process of knowing how the AI black box performs [150,151]. The origination of evidence for scientific validation is also nowadays coined under the umbrella of explainable AI because XAI is used for satisfying the evidence for scientific validation [15,123,128,152,153,154]. Recently, explainable artificial intelligence (XAI) has attracted much interest in medicine. To reach a level of explainable medicine we need causability, where causability encompasses measurements for the quality of explanations, in the same way, that usability encompasses measurements for the quality of use [155,156,157].

### 7.5. Strength, Weakness, and Extension

Radiomics and radiogenomics are the next generations of methodologies for non-invasive cancer treatments, which consider the radiological features and molecular characteristics of the disease. The quantitative imaging feature (phenotype or radiomics) describes the disease characteristics in the case of radiomics analysis. Similarly, both the phenotype and genotype of the cancer are useful for radiogenomics analysis. However, artificial intelligence frameworks such as ML, DL, and TL have provided an edge over traditional analysis of the disease. They have many positive outcomes such as survival prediction, tumor grading, imaging biomarker, clinical decision making, treatment planning, risk stratification, personalized treatment, and precision medicine. As it provides an entirely non-invasive diagnosis with an effective outcome, it is the most considered and most rated treatment choice over invasive treatment methods such as biopsy, surgery, chemotherapy, and radiation therapy.

Over the past few years, numerous research analyses have been carried out on this emerging technology; however, systematic computation is still a matter of concern due to several factors. The scarcity and the lack of availability of standard and volumetrics radiogenomics public datasets may lead to poor training of the AI models of ML and DL. The training can be improved by breeding, fusion, or stochastic imaging methods with AI models for superior segmentation paradigms. These models are data-centric and provide better accuracy on the test dataset when trained with volumetric datasets. Additionally, multi-regional, inter-, and intra-institutional heterogeneity datasets are needed as the genetic behaviors of the different demographic areas vary. Gene expression and signaling pathways are exceptionally complex, therefore it is tiresome work to generate a huge quantity of data from whole-genome sequencing with imaging data. As we observed from the statistical analysis, most of the current research on radiogenomics has been conducted in the USA; hence, other parts of the globe are still unable to share its view. Hence, there should be a need for the involvement of multi-center studies. Examples of multi-center studies can be seen here for other applications [34,38,158]. Another extension would be to use extensive data analysis for the radiogenomics [159]. 

Furthermore, one could study the brain connectivity in brain tumor patients [160,161]. Association studies can be incorporated, which study the relationships between brain tumor radiomics and genomics for survival analysis [129,162]. Radiogenomics is inevitable and needs further high-level, multi-center, multi-intuitional, multi-geographical research. The standardized workflow of radiogenomics studies needs a computational guide to provide a different level of performance. Lastly, there is the opportunity of studying the effect of COVID-19 on brain tumor patients [163].

## 8. Conclusions

A brain tumor is a deadly disease where several biological assessments and biomedical research studies have been conducted to understand the tumorigenesis over the decade. Brain tumor characterization is the process of learning the molecular signature of the underlying nature of the disease in terms of lesion segmentation, classification, detection, and analysis. Radiomics and radiogenomics are relatively new fields of research on cancer that provide hope for the most acceptable assessment methods. Radiogenomics is the association between established MR imaging features and the molecular characteristics of brain tumors to improve disease diagnosis and characterization with greater precision. Radiomics and radiogenomics are promising fields in cancer research treatment providing personalized treatment planning, precision medicine, and imaging biomarkers during different stages of treatment of the patients. These technologies are not intended to replace the radiologist completely; however, they need to accept this new technology and make the necessary adjustments to accommodate the changes it will bring to the clinical process.

## Figures and Tables

**Figure 1 cancers-14-04052-f001:**
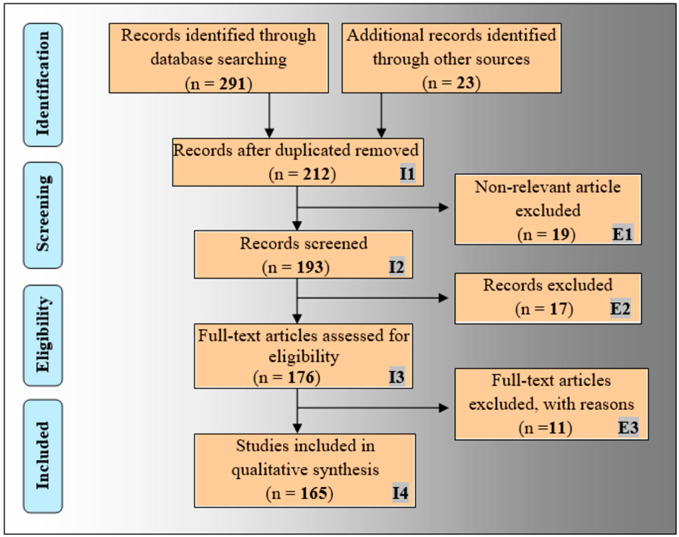
The PRISMA framework for the flow diagram of the selection process.

**Figure 2 cancers-14-04052-f002:**
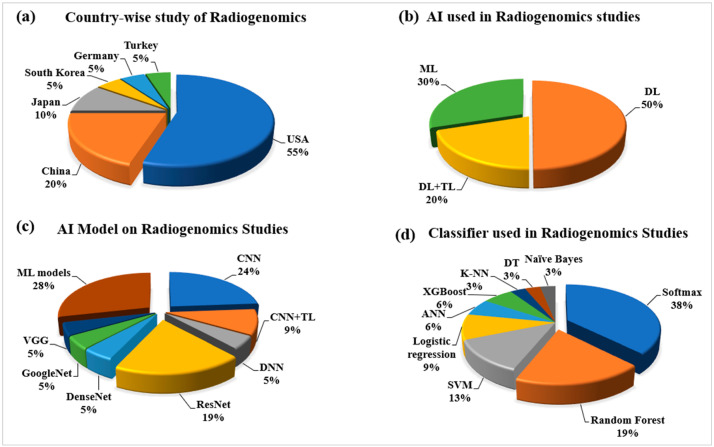
Statistical distribution for radiogenomics studies: (**a**) country-wise; (**b**) types of AI technology; (**c**) types of AI models; and (**d**) AI-based classifiers used in radiogenomics. Notes: ML: machine learning, DL: deep learning, TL: transfer learning, CNN: convolutional neural network, DNN: deep neural network, VGG: visual geometric group, SVM: support vector machine, ANN: artificial neural network, K-NN: K-nearest neighbor, DT: decision tree.

**Figure 3 cancers-14-04052-f003:**
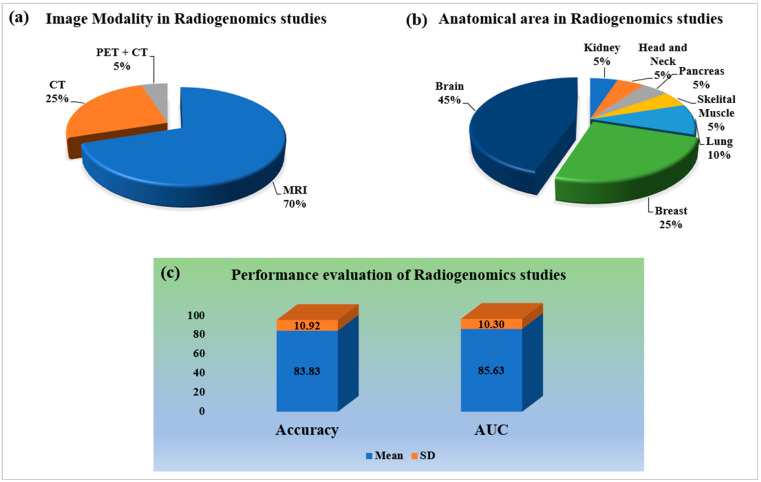
Statistical distribution for radiogenomics studies: (**a**) imaging modality; (**b**) anatomical area; (**c**) performance evaluation. Notes: CT: computer tomography, PET: positron emission tomography, MRI: magnetic resonance imaging, AUC: area under curve, SD: standard deviation.

**Figure 4 cancers-14-04052-f004:**
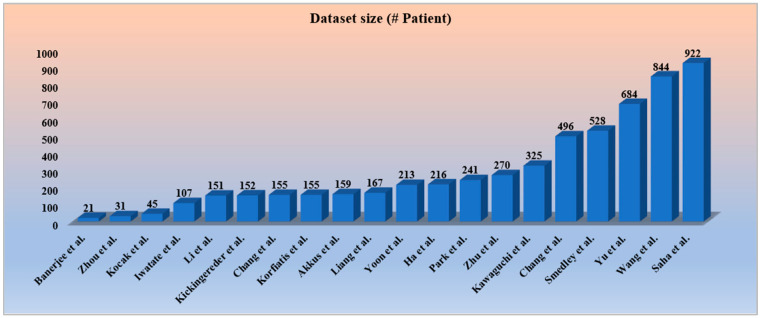
The distribution of increasing dataset size in various radiogenomics studies [53,54,55,56,57,58,59,60,61,62,63,64,65,66,67,68,69,70,71].

**Figure 5 cancers-14-04052-f005:**
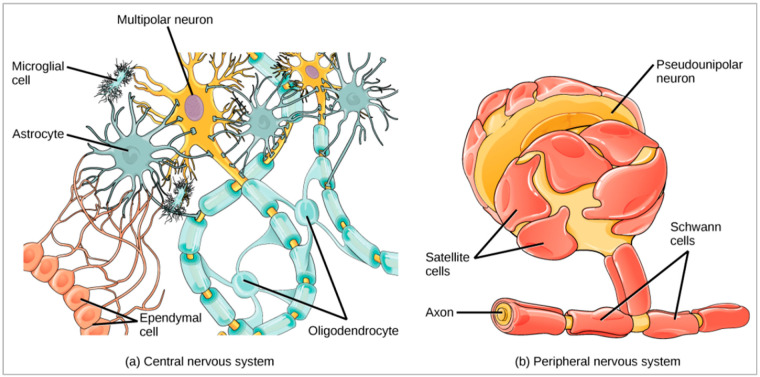
Glia cells of the neuron and their environment of the central nervous system and peripheral nervous system [88].

**Figure 6 cancers-14-04052-f006:**
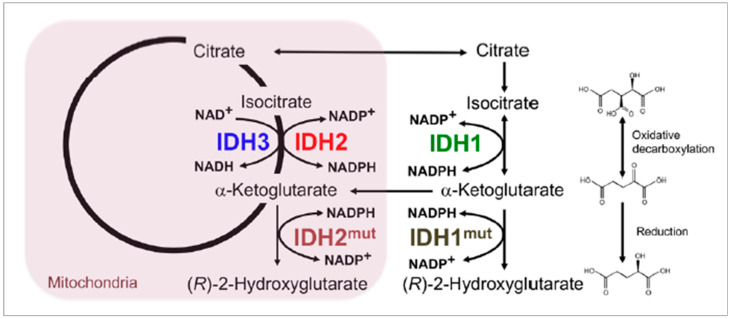
Chemical reaction and metabolic pathways present in a brain tumor cell, with emphasis on enzymatic effectors—IDH1 and IDH2 mutations [92].

**Figure 7 cancers-14-04052-f007:**
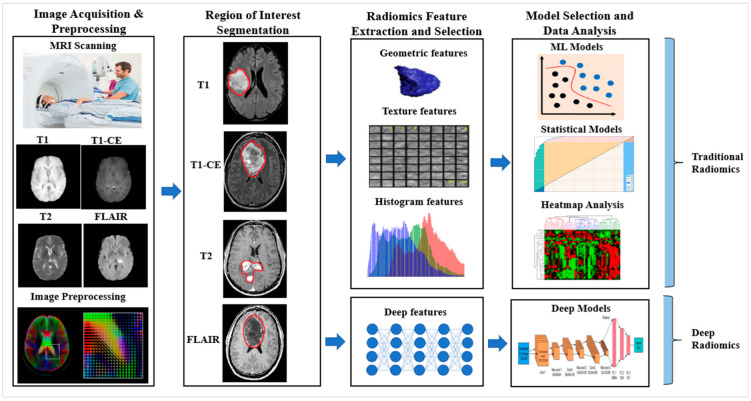
Radiomics workflow for brain lesion characterization. Notes: T1: T1-weighted MRI, T2: T2-weighted MRI, T1-CE: T1-contrast-enhanced, FLAIR: fluid-attenuated inversion recovery.

**Figure 8 cancers-14-04052-f008:**
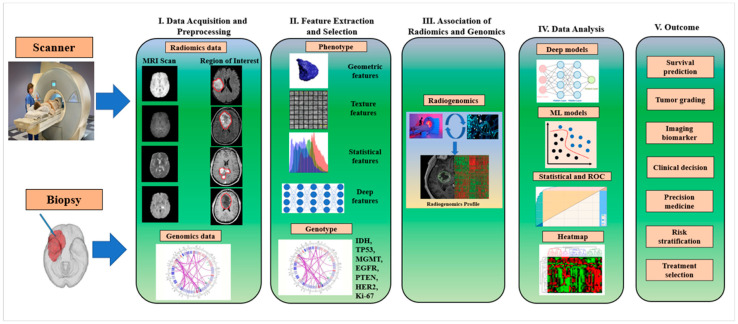
The workflow of radiogenomics for brain tumor genomics and disease characterization. Notes: IDH: isocitrate dehydrogenase, TP53: tumor protein53, MGMT: O6-methylguanine DNA methyltransferase, EGFR: epidermal growth factor receptor, PTEN: phosphatase and tensin homolog, HER2: human epidermal growth factor receptor 2.

**Figure 9 cancers-14-04052-f009:**
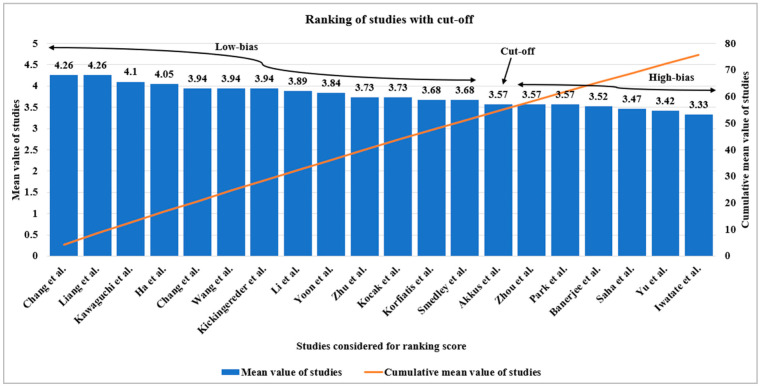
The ranking score technique shows the frequency distribution of radiogenomics studies in descending order succeeded by the cumulative plot, showing the raw cut-off mark for bias analysis [51,52,53,54,55,56,57,58,59,60,61,62,63,64,65,66,67,68,69,70,71].

**Table 1 cancers-14-04052-t001:** WHO grading of CNS and brain tumors.

Grading	Some Selected Types of CNS and Brain Tumor
Grade 1	Meningioma, solitary fibrous tumor, diffuse astrocytoma (*MYB*- or *MYBL1*-altered), polymorphous low-grade neuroepithelial tumor of the young, multi-nodular and vacuolating neuronal tumor.
Grade 2	Meningioma, solitary fibrous tumor, oligodendroglioma, astrocytoma (IDH-mutant), IDH-mutant, and 1p/19q-co-deleted, myxopapillary ependymoma, pleomorphic xanthoastrocytoma, supratentorial ependymoma, posterior fossa ependymoma.
Grade 3	Meningioma, solitary fibrous tumor, oligodendroglioma, Astrocytoma (IDH-mutant), IDH-mutant, and 1p/19q-co-deleted, pleomorphic xanthoastrocytoma, supratentorial ependymoma posterior fossa ependymoma.
Grade 4	Glioblastoma (IDH-wildtype), astrocytoma (IDH-mutant), diffuse hemispheric glioma (H3 G34-mutant).

**Table 2 cancers-14-04052-t002:** Brain tumor types and their underlying genetic and epigenetic alterations [33,52,89,90,106,107,108].

SN.	Brain and CNS Tumor Types	Key Genes and Protein Alterations for Tumor
1	Astrocytoma Grade I: Pilocytic Astrocytoma	BRAF, NF1, KIAA1549-BRAF
2	Astrocytoma Grade II: Low-grade Astrocytoma	EGFR1, BRAF
3	Astrocytoma Grade III: Anaplastic Astrocytoma	IDH1/2, TP53, ATRX, CDKN2A/B
4	Astrocytoma Grade IV: Glioblastoma (GBM)	IDH1/2, TERT, chromosomes 7/10, EGFR
5	Oligodendroglioma	IDH1/2, TERT promoter, 1p/19q, NOTCH1, FUBP1, CIC
6	Angiocentric glioma	MYB
7	Diffuse astrocytoma	MYB, MYBL1
8	Medulloblastoma	TP53, CTNNB1, PTCH1, APC, SUFU, GLI2, SMO, MYC, MYCN, PRDM6, KDM6A.
9	Meningiomas	NF2, TRAF7, AKT1, PIK3CA; SMO, SMARCE1, KLF4, BAP1 in subtypes; H3K27me3; TERT, CDKN2A/B in CNS WHO grade 3
10	Retinoblastoma	Retinoblastoma (Rb) protein
11	Ependymomas	ZFTA, YAP1, RELA, MAML2, H3 K27me3, NF1, NF2, EZHIP, MYCN, KMT2D, RELA, FANCE, and EP300
12	Primitive neuroectodermal tumors	Isochrome (17q)
13	Astroblastoma	MN1
14	Chordoid glioma	PRKCA
15	Ganglion cell tumors	BRAF
16	Polymorphous low-grade neuroepithelial tumor	BRAF, FGFR family
17	Diffuse midline glioma, H3 K27-altered	TP53, H3 K27, PDGFRA, EGFR, ACVR1, EZHIP
18	Diffuse hemispheric glioma, H3 G34-mutant	TP53, H3 G34, ATRX
19	Diffuse pediatric-type high-grade glioma	IDH-wildtype, H3-wildtype, MYCN, PDGFRA, EGFR
20	Infant-type hemispheric glioma	NTRK family, ROS, ALK, MET
21	High-grade astrocytoma with piloid features	ATRX, BRAF, CDKN2A/B (methylome), NF1
22	Pleomorphic xanthoastrocytoma	CDKN2A/B, BRAF
23	Subependymal giant cell astrocytoma	TSC1, TSC2
24	Solitary fibrous tumor	NAB2-STAT6
25	Meningeal melanocytic tumors	NRAS (diffuse), GNA11, GNAQ, CYSLTR2, PLCB4
26	Atypical teratoid/rhabdoid tumor	SMARCA4, SMARCB1
27	Embryonal tumor with multi-layered rosettes	C19MC, DICER1
28	Glioneuronal tumor	NF1, PDFGRA, PRKCA, FGFR1, PIK3CA, KIAA1549-BRAF fusion, 1p, Chromosome 14
29	Dysplastic cerebellar gangliocytoma	PTEN
30	Extraventricular neurocytoma	IDH-wildtype, FGFR (FGFR1-TACC1 fusion)
31	Multi-nodular and vacuolating neuronal tumor	MAPK pathway
32	Dysembryoplastic neuroepithelial tumor	FGFR1
33	CNS neuroblastoma	FOXR2, BCOR
34	Desmoplastic myxoid tumor of the pineal region	SMARCB1

BRAF: proto-oncogene B-Raf, NF1: neurofibromin 1, FGER: fibroblast growth factor receptors, EGFR: epidermal growth factor receptor, IDH: isocitrate dehydrogenase, ATRX: alpha thalassemia X-linked mental retardation, TP53: tumor protein53, CDKN2A/B: cyclin-dependent kinase inhibitor 2A/B, TERT: telomerase reverse transcriptase, CIC: capicua transcriptional repressor, FUBP1: far upstream element binding protein 1, NOTCH1: notch homolog 1, MYB: myeloblastosis, MYBL1: MYB like1, CTNNB1: catenin beta-1, APC: adenomatous polyposis coli, PTCH1: protein patched homolog 1, SUFU: suppressor of fused protein, SMO: smoothened, AKT1: threonine kinase 1, TRAF7: TNF receptor associated factor 7, ZFTA: zinc finger translocation associated, YAP1: yes-associated protein 1, MAML2: mastermind-like transcriptional coactivator 2, H3: histone3, EZHIP: EZH inhibitory protein, KMT2D: lysine methyltransferase 2D, FANCE: FA complementation group E, MN1: meningioma, PRKCA: protein kinase C alpha, ACVR1: activin A receptor type 1, PDGFRA: platelet-derived growth factor receptor alpha, *NTRK*: neurotrophic tyrosine receptor kinase, *ALK*: anaplastic lymphoma kinase, MET: mesenchymal epithelial transition, TSC: tuberous sclerosis, STAT6: signal transducer and activator of transcription 6, NRAS: neuroblastoma RAS viral oncogene homolog, GNAQ: G protein subunit alpha Q, GNA11: G protein subunit alpha 11, PLCB4: phospholipase C beta 4, CYSLTR2: cysteinyl leukotriene receptor 2, SMARCB: SWI/SNF related, matrix associated, actin dependent regulator Of chromatin, subfamily B C19MC: chromosome 19 miRNA cluster, DICER1: dicer 1, ribonuclease III, PRKCA: protein kinase C alpha, PTEN: phosphatase tensin homologue, MAPK: mitogen-activated protein kinase, FOXR2: forkhead Box R2, BCOR: BCL6 corepressor.

**Table 3 cancers-14-04052-t003:** Comparison among the image modalities.

Factor	MRI	CT	X-Ray	Ultrasound
Duration	30–45 min	3–7 min	2–3 min	5–10 min
Cost	High	Moderate	Low	Low
Soft tissue	Excellent detail	Poor detail	Poor detail	Poor detail
Bone	Poor detail	Excellent detail	Excellent detail	Poor detail
Dimension	3	3	2	2
Radiation	No	10 mSv	0.15 mSv	No

Note: mSv: *millisievert.*

**Table 4 cancers-14-04052-t004:** AI model for radiogenomics study on the brain tumor.

Author, Year and Reference	Image Modality	Radiomics Feature	Genomics Feature	AI Model Used	Result
Akkus et al. [53] (2016)	MRI: T1-CE, T2	Deep radiomics	1p19q deletion of LGG	DL (CNN)	Acc.: 87.7
Kickingereder et al. [64] (2016)	MRI: T1, T1-CE, FLAIR, DWI, DSWCEI, PSWI	Hand-crafted	EGFR, PTEN, PDGFRA, MDM4, CDK4 CDKN2A, NF1, and RB1	ML	Acc.: 63 AUC: 69
Chang et al. [60] (2017)	MRI: T1, T1-CE, T2, FLAIR	Deep radiomics	IDH1 prediction for LGG	DL (ResNet)	Acc.: 89.1 AUC: 95
Li et al. [66] (2017)	MRI: T1, T2	Deep radiomics	IDH1 prediction for LGG	DL (CNN)	Acc.: 92.4 AUC: 95
Liang et al. [67] (2017)	MRI: T1, T1-CE, T2, FLAIR	Deep radiomics	IDH1 prediction for Glioma	DL (DenseNet)	Acc.: 91.4 AUC: 94.8
Korfiatis et al. [65] (2017)	MRI: T2	Deep radiomics	MGMT status	DL (ResNet50)	Acc.: 94.9
Chang et al. [61] (2018)	MRI: T1, FLAIR	Deep radiomics	IDH1, 1p/19q co-deletion, MGMT	DL (ResNet)	Acc.: 94 AUC: 91
Smedley et al. [68] (2018)	MRI: T1-CE, T2, FLAIR	Deep radiomics	Tumor morphology	DL (AE)	MAE: 0.0114
Calabrese et al. [131] (2020)	MRI: T1, T1-CE, T2, FLAIR, SWI, DWI, ASLPI, HARDI	Deep radiomics	ATRX, IDH, 7/10aneuploidy, CDKN2, EGFR, TERT, PTEN, TP53, MGMT	TL (CNN+ RF)	AUC: 97
Kawaguchi et al. [63] (2021)	MRI: T1, T1-CE, T2, FLAIR	Hand-crafted	IDH, MGMT, TERT, 1p19q	ML	AUC: 90

Abbreviation: DWI: diffusion-weighted image, SWI: susceptibility-weighted image, DSWCEI: dynamic susceptibility-weighted contrast-enhanced imaging, PSWI: pre-contrast susceptibility-weighted imaging, ASLPI: arterial spin labeling perfusion images, HARDI: high angular resolution diffusion imaging, Acc: accuracy, AUC: area under ROC curve, MAE: mean absolute error, AE: auto-encoder, RF: random forest.

**Table 5 cancers-14-04052-t005:** Benchmarking table.

Author, Year and Reference	Radiomics	Radio- Genomics	AI Framework	Anatomical Cancer Discussed	Statistics and Risk-of-Bias (RoB) Analysis
Rizzo et al. (2018) [114]	✓	🗶	🗶	Generalized	🗶
Kazerooni et al. (2019) [135]	🗶	✓	🗶	Brain (Glioblastoma)	🗶
Bodalal et al. (2019) [146]	🗶	✓	✓	All Cancer	🗶
Trivizakis et al. (2020) [51]	🗶	✓	✓	All Cancer	🗶
Gullo et al. (2020) [139]	🗶	✓	🗶	All Cancer	Statistical analysis only
Shui et al. (2021) [147]	🗶	✓	✓	All Cancer	🗶
Singh et al. (2021) [148]	✓	✓	🗶	Brain (Glioma)	🗶
Wu et al. (2021) [113]	✓	🗶	✓	Lung	🗶
Habib et al. (2021) [49]	✓	✓	🗶	Brain	🗶
Jena et al. (2022) (Proposed)	✓	✓	✓	Brain	✓

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
