# Peer review of "Brain Tumor Characterization Using Radiogenomics in Artificial Intelligence Framework"

_cancers, 2022, doi:10.3390/cancers14164052_

Round 1

Reviewer 1 Report

Thank you for your submission.

Authors wrote about brain tumor characterization.

The manuscript is worthwhile to publish.

I think there is only one problem. The graph layout should be modified.

Author Response

Thank you for your constructive and positive feedback on our work. 

Reviewer 2 Report

Dear authors of the manuscript on "Brain Tumor Characterization using Radiogenomics in Artificial Intelligence Framework",
your manuscript presents an overview on the current topics of radiogenomics. As for now, it seems not to fit in as suitable publication and should be thoroughly refined:

* If you use figures from other publications, please refine and add new insights to them! Therefore, figures 5-7 have to be changed. There is even a copyright on Figure 6?!
* there is no common thread
* chapter three, after presenting results before, seems out of scope and is merely an introduction to basics of brain tumor biology, except for 3.3 as introduction to classification could be integrated in the storyline. Still, this chapter should summarize recent advances in related classification approaches in depth.
* same again for chapter 4, there are other reviews on this topic  that can be cited instead of giving superficial information on some relevant biomarkers (while leaving out relevant information on others as this topic is now presented).
* and equally true for intro of 5, could be better incorporated into the main "story" and summarized
The most relevant information is given in chapter 6 and could be extended!
* Subsections of 6 and (7)Discussion could also be enumerated
* Table 2: missing references to the data provided
* Figure legends, please provide information on abbreviations
* Dataset overview is incomplete.
* Regarding the identification of studies via databases and registers, check add additional works on VASARI and REMBRANDT.
* Comment: fostering open data in cancer is of course important, additionally, public datasets will be even more scarce in future, one could include ways how to handle controlled data as well as incorporate privacy-preserving methods.
* Are there any insights into age differences?
* Line 214 where is here?
* mutation vs mutation process?
* minor typos such as ITLAY vs ITALY, missing punchlines...
All the best and continued success with your revision and ongoing research!

Author Response

Thank you for the suggestions.

Reviewer 3 Report

I enjoyed reading this review manuscript. The radiogenomic approaches to characterize brain tumors are latest and exciting topics in this field. This review paper is worthwhile to be published in Cancers for broad range of audiences especially oncologists. 

I have minor comments regarding figure arrangement. Figure 8 and Figure 9 are busy figures. Authors are encouraged to revise these figures clearly.  

Author Response

(The authors gave the same response as above.)

Round 2

Reviewer 2 Report

Dear authors,

You mention to have used images without copyright issues, and I am fully aware that one could obtain the rights for copied images, still, I cannot completely agree with your response and would recommend to create refined figures incorporating some features of this review's focus. Otherwise, images as Figure 6 are at least not of interest or not "necessary" to the target audience! Moreover, a statement like "figure taken from" and as soon as it is refined, "adapted from" should be included. I agree that basics from several disciplines have to be introduced, still, authors are encouraged to conentrate on the key messages of this review!

Additional comments:
* Please add the reference to p14l452.
* What does the "here" in the 1st introductory sentence in paragraph 3 mean "biology or pathophysiology of brain tumors has been discussed here" -> please add the respective references!
* Please add the respective reference(s) to the paragraph on "The MRI has several limitations.
* Reference 161 is questionable, readers may need some more references for XAI and could be interested in the term causability.
* Why did you not use brain tumor as another keyword in the first identificatiion step?
* and clarify, what do the 2 exclusion criteria "no-relevant" and "insufficient data" mean in more detail!
* last, please again conduct proof reading for readability (f.i. unnecessary preoposition "the" XAI)

Author Response

Rebuttal

14th August 2022

Reviewer #2

Comments and Suggestions for Authors

2.1 You mention to have used images without copyright issues, and I am fully aware that one could obtain the rights for copied images, still, I cannot completely agree with your response and would recommend to create refined figures incorporating some features of this review's focus. Otherwise, images as Figure 6 are at least not of interest or not "necessary" to the target audience!

Moreover, a statement like "figure taken from" and as soon as it is refined, "adapted from" should be included. I agree that basics from several disciplines have to be introduced, still, authors are encouraged to concentrate on the key messages of this review!

Author: Thank you for the suggestions.  Yes, we wholeheartedly and completely agree with the reviewer on the concern of adapting images from other sources. Therefore, the issues raised in Figures 5,6, and 7 are clarified below.

Figure 5: For Figure 5, is adapted from a source, that provides “free to use” without copyright issues.

Source: https://openstax.org/books/biology/pages/35-1-neurons-and-glial-cells

(Provided by: OpenStax. Located at: https://openstax.org/books/biology/pages/35-1-neurons-and-glial-cells, LicenseCC BY: AttributionLicense Terms: “Access for free” at https://openstax.org/books/biology/pages/35-1-neurons-and-glial-cells).

Thank you for understanding the usage of Figure #5.

Figure 6: We have removed Figure 6 as the suggestion from the reviewer, as this is not of interest or not "necessary" to the target audience. As it very general figure depicting the mutation process, we also completely agree with the reviewer. Thank you once again for figure #6.

Figure 7: We have replaced Figure 7, with another image with the same theme and this new figure is also “free from copyright issues”.

Source: https://www.science.org/doi/10.1126/sciadv.aaw4543

(Copyright © 2019 The Authors, some rights reserved; exclusive licensee American Association for the Advancement of Science. No claim to original U.S. Government Works. Distributed under a Creative Commons Attribution-NonCommercial License 4.0 (CC BY-NC).

This is an open-access article distributed under the terms of the Creative Commons Attribution-NonCommercial license, which permits use, distribution, and reproduction in any medium, so long as the resultant use is not for commercial advantage provided the original work is properly cited.)

We have now fully resolved the issue regarding the copyright for Figures 5, and 7 while removing figure #6. Thank you for the support in improving the readability of the manuscript.

Additional comments

2.2 Please add the reference to p14l452.

Author: Thank you for the suggestions.  We have added new references in the revised manuscript on page #14 line # 505. Thank you once again for improving the readability of the manuscript.

2.3 What does the "here" in the 1st introductory sentence in paragraph 3 mean "biology or pathophysiology of brain tumors has been discussed here" -> please add the respective references!

Author: Thank you for the suggestions.  The word “here” stands for “in this section”, which we have now replaced in the revised version of the manuscript.

Thank you once again for correcting us and improving the readability of the manuscript.

2.4 Please add the respective reference(s) to the paragraph on "The MRI has several limitations.

Author: Thank you for the suggestions.  We have added relevant references in support of the statement “The MRI has several limitations”, in the revised version of the manuscript on page #14, line 514. Thank you once again for correcting us and improving the readability of the manuscript.

2.5 Reference 161 is questionable, readers may need some more references for XAI and could be interested in the term causability.

Author: Thank you for the suggestions.  Again, as per the suggestions, we have added the supporting reference for causability of explainable AI (XAI), which is reflected in the revised manuscript with a highlighted yellow mark on page #23.

2.6 Why did you not use brain tumor as another keyword in the first identification step?

Author: Thank you for the suggestions. We greatly appreciate it.  We have added the new keyword “brain tumor” in the first identification step in the keyword list.  

Thank you once again for improving the readability of the manuscript.

2.7 and clarify, what do the 2 exclusion criteria "no-relevant" and "insufficient data" mean in more detail!

Author: Thank you for the suggestions.  The exclusion criteria “no-relevant” suggest that some articles which are not relevant to brain tumor characterization using radiogenomics are excluded from the collection such as they may be related to other cancer. Whereas the “insufficient data” suggest that the publications are satisfying the criteria of our subject “brain tumor characterization using radiogenomics”, however, the insufficient data may be observed in terms of dataset size, no use of all required procedures or protocols, or even the algorithm details in radiogenomics.

Thank you once again for correcting us and improving the readability of the manuscript.

2.8 last, please again conduct proof reading for readability (f.i. unnecessary proposition "the" XAI)

Author: Thank you for the suggestions. We have performed rigorous proofreading of the whole manuscript once again to remove the typographical, stylistic, and grammar errors. Further, one of our very senior authors further examined the manuscript word-by-word to ensure the flow, grammar, connectedness, and readability are to the highest level.

We have checked our grammar errors on the Grammarly tool as well. Thank you once again for improving the readability of the manuscript.

Round 3

Reviewer 2 Report

Dear authors,

thank you for changing images, still, I always recommend to create own/novel images with focus on the main topic, or adaptions instead of copies that only relate to and do not cover the content described in the manuscript.

The keyword "brain tumor" for the identification step, in sense of the review search process, and not the keyword list of the manuscript, depicts an additional condition to the search strategy in respect of this review's focus on brain tumors. For now, your search covers other medical/biological areas as well, which should be discussed if not previously limited by the search strategy.

Best regards for your continued work!

Author Response

Rebuttal

17th August 2022

Reviewer #2

Comments and Suggestions for Authors

Dear authors,

thank you for changing images, still, I always recommend to create own/novel images with focus on the main topic, or adaptions instead of copies that only relate to and do not cover the content described in the manuscript.

Author: Thank you for your recommendation and precious advice regarding copyright issues. We will adhere to the recommendations for all our future projects. We greatly appreciate the time that you have taken to review our manuscript and provide insightful feedback. Once again thank you for your continuous support, evaluation, recommendation, and advice for increasing the readability of the manuscript.

2.1 The keyword "brain tumor" for the identification step, in sense of the review search process, and not the keyword list of the manuscript, depicts an additional condition to the search strategy in respect of this review's focus on brain tumors. For now, your search covers other medical/biological areas as well, which should be discussed if not previously limited by the search strategy.

Author: Thank you for the comment. We are really regretful for the misinterpretation. We have added the keyword “brain tumor” for the identification step in the review search process or search strategy. The modification is reflected in the revised manuscript on page #3. Once again thank you for correcting us and enhancing the readability of our manuscript.

2.2 Best regards for your continued work!

Author: Thank you for your kind words and your continued support throughout the evaluation.
